# Development of an Ultrasensitive ELISA Assay for Evaluating HIV-1 Envelope Glycoprotein as a Marker for Targeted Activator of Cell Kill

**DOI:** 10.3390/v18010046

**Published:** 2025-12-28

**Authors:** Guoxin Wu, Luca Sardo, Yuan Li, Steven D. Heron, BaoJen Shyong, Matthew Mazur, Daniel M. Gorman, Carl J. Balibar, Brian C. Magliaro, Craig Leach, Thomas Rush, Bonnie J. Howell

**Affiliations:** 1MRL, Merck & Co., Inc., Rahway, NJ 07065, USA; 2Eurofins Professional Scientific Services Mid Atlantic, LLC, Lancaster, PA 17601, USA

**Keywords:** HIV-1, gp120, antibody, ELISA, Quanterix, immunocytochemistry, mass spectrometry (MS), TACK, biomarker

## Abstract

The HIV-1 envelope glycoprotein gp120 is prominently exposed on the surface of both HIV-1 virions and infected host cells, serving as a key marker of infection. gp120 plays a pivotal role in viral entry by interacting with the primary receptor, CD4, on host cells. Therapeutic strategies targeting the HIV-1 reservoir, such as anti-gp120 antibodies that trigger antibody-dependent cellular cytotoxicity (ADCC) and chimeric antigen receptor T (CAR-T) cells, rely on the presence of gp120 on the surface of infected cells to exert their effects. Consequently, accurate monitoring of gp120 expression on infected cells is essential for evaluating the pharmacological efficacy of these interventions. In this study, a sensitive, specific, and inexpensive enzyme-linked immunosorbent assay (ELISA) for quantifying HIV-1 gp120 glycoprotein was developed using a selected pair of anti-gp120 antibodies. The assay achieved a lower limit of quantitation (LLOQ) of 0.16 pM, demonstrating sensitivity comparable to that of the digital single molecule array (Simoa) platform, which exhibited a LLOQ of 0.23 pM and requires specialized instrumentation. The binding specificity of the antibodies used in the novel assay was confirmed using liquid chromatography–mass spectrometry (LC-MS), and the assay was pharmacologically validated with lysates obtained from 2D10 and MOLT IIIB cell lines. Furthermore, treatment of HIV-infected human primary CD4^+^ T cells with a targeted activator of cell kill (TACK) compound significantly reduced gp120 concentration in CD4^+^ T cell lysate compared to controls. The gp120 marker from infected cell lysates correlated with the number of gp120-positive cells detected by immunocytochemistry, as well as with HIV-1 p24 levels and cell-associated viral RNA measurements. In summary, a novel, simple, and sensitive HIV-1 gp120 ELISA has been developed and validated. This assay holds potential for investigating HIV-1 persistence and evaluating the efficacy of therapeutic agents targeting infected cells.

## 1. Introduction

Over the past several decades, significant progress in HIV-1 treatment has been achieved through the development of potent combination antiretroviral therapy (cART). While cART effectively suppresses viral replication and limits cell-to-cell transmission, it does not eliminate virus that has already integrated into host cells. Discontinuation of cART inevitably leads to viral rebound due to the persistence of HIV-1 reservoirs [1,2,3]. Achieving either treatment-free remission or complete viral eradication by targeting and eliminating the HIV-1 reservoir is critical for attaining an effective HIV-1 cure [4].

Targeting the HIV-1 envelope glycoprotein gp120 has emerged as a promising strategy in efforts to cure HIV-1. gp120 is expressed on the surface of HIV-1 virions and infected cells and plays a key role in viral entry by directly interacting with the CD4 receptor on host T lymphocytes and triggering a cascade of events that lead to the fusion of the viral and host cell membranes [5,6].

Several immunoassays for quantifying HIV-1 gp120 have been reported [7,8,9,10]. These methods, with detection sensitivities ranging between 3 to 130 pM, are primarily suited for detecting gp120 in plasma and cell lysates derived from viremic individuals living with HIV-1. However, they lack the sensitivity required for measuring gp120 in samples from individuals with suppressed viremia. In addition, some sensitive platforms are costly and require specialized instrumentation. Therefore, developing inexpensive higher-sensitivity molecular methods to accurately quantify gp120 expression in infected cells and tissues is essential for evaluating target engagement and assessing the efficacy of antibody-dependent cell-mediated cytotoxicity (ADCC) [11].

In this study, we developed a specific and highly sensitive ELISA method for quantifying HIV-1 gp120 by screening a comprehensive panel of anti-gp120 antibodies and evaluating paired combinations for their ability to capture and detect gp120. The optimal antibodies were integrated into digital ELISA using the single molecule array (Simoa) technology and liquid chromatography–mass spectrometry (LC-MS), to determine the detection limits and performance characteristics. Our method was further evaluated for its ability to assess the effect of targeted activator of cell kill (TACK) molecules in eliminating HIV-1-infected human CD4^+^ T cells. This novel ELISA demonstrates that gp120 envelope glycoprotein holds promise as an alternative biomarker for studying HIV-1 persistence and for evaluating therapeutic interventions aimed at achieving an HIV-1 cure.

## 2. Materials and Methods

### 2.1. Reagents and Cells

Recombinant biotinylated human soluble CD4 (sCD4) protein (Cat# CD4-H82E8) was obtained from ACRO Biosystems (Newark, DE, USA), reconstituted and stored as recommended. Recombinant HIV-1 IIIB envelope gp120 protein expressed in HEK293 (Cat# ab174070) was purchased from Abcam (Waltham, MA, USA). Recombinant full-length HIV-1 IIIB gp160 protein (Cat# MBS5304450) expressed in baculovirus was obtained from MyBiosource (San Diego, CA, USA). A panel of 11 anti-gp120 antibodies were evaluated in this study, with source information provided in Table 1. Three selected anti-gp120 antibodies, including clones VRC07, 447-52D, and 2G12, were labeled in-house with alkaline phosphatase (AP) according to a previously described procedure [12]. The labeled antibodies were stored at -20 °C in 1× PBS buffer containing 50% glycerol. Phorbol 12-myristate 13-acetate (PMA), ionomycin, and tumor necrosis factor alpha (TNF-α) were obtained from Millipore-Sigma (St. Louis, MO, USA). The epigenetic histone deacetylase inhibitor (HDACi) vorinostat (VOR) was synthesized in-house. The 2D10 cell line, stably transfected with inducible HIV-1 full length envelope protein tagged with green fluorescent protein (GFP), was obtained from the laboratory of Jonathan Karn [13]. The MOLT IIIB cell line, containing integrated HIV-1 provirus (clade B), was obtained from the NIH [14]. HIV isolates 93RQ034, JRFL B, and 93MW959, which represent HIV clade A, B, and C, respectively, were obtained from the NIH AIDS Reference and Reagent Repository (Germantown, MD, USA) as previously described [15].

### 2.2. gp120 ELISA

Four distinct ELISA platforms were developed for detecting gp120. First, a direct ELISA was implemented. For this purpose, recombinant purified HIV-1 envelope proteins, including gp120 and gp160, were coated onto distinct high-binding microtiter ELISA plates (Cat#3925, Corning, NY, USA) at 0.1 µg/mL and detected using either biotin-labeled soluble CD4 (sCD4) protein followed by streptavidin-AP or AP-labeled anti-gp120 antibody VRC07 [16,17]. Second, an indirect ELISA was used for selecting the best anti-gp120 binding antibodies by coating gp120 protein onto the plates, adding primary anti-gp120 antibodies (0.1 µg/mL each), and detecting them with either an AP-labeled secondary goat anti-mouse, or an anti-rabbit, or an anti-human IgG antibody. Next, a sandwich ELISA was performed to evaluate the gp120 capture reagent by coating ELISA plates with either sCD4 protein or the anti-gp120 antibody 2G12 [18], adding gp120 protein, followed by detection with the AP-labeled antibody VRC07. After that, the best gp120 sandwich ELISA antibody pair was identified by coating plates with different anti-gp120 antibodies, adding gp120 protein, then detecting the captured gp120 with various AP-labeled anti-gp120 antibodies. Assay plates for gp120 sandwich ELISA were incubated overnight at 4°C with shaking at 30 rpm. Following incubation, the plates were washed six times with phosphate-buffered saline containing 0.05% Tween-20 (PBST), and 100 µL of AP enzyme substrate (Cat# T-2214, Thermo Fisher, Carlsbad, CA, USA) was added. Plates were incubated at room temperature for 30 min, and luminescent signals were read using an Envision plate reader (Revvity, Waltham, MA, USA) [19].

### 2.3. Flow Cytometry

The GFP-tagged HIV-1 envelope protein expressed in the 2D10 cell line was detected using flow cytometry as previously described [20]. Cultured 2D10 cells were treated with various concentrations of PMA and ionomycin starting from 100 ng/mL PMA/1 µg/mL ionomycin, followed by 10-fold serial dilutions out to 0.01 ng PMA/0.1 ng/mL ionomycin, or left untreated for 24 h. Treated cells were collected, counted, and divided into two groups. In the first group, cells were fixed with cytofix/cytoperm buffer from BD Biosciences (San Diego, CA, USA) for 30 min, washed twice with 1 mL of 1% BSA/PBS, and resuspended in PBS at 2 × 10^6^/mL. GFP signal was analyzed using a FACSymphony A3 cytometer from BD Biosciences (San Diego, CA, USA). In the second group, cells were pelleted for subsequent gp120 ELISA.

### 2.4. gp120 Single Molecule Array (Simoa)

The detection sensitivity of the gp120 ELISA was compared with that of the Quanterix digital ELISA Simoa platform, employing the same antibody pair (2G12 and VRC07). VRC07 was labeled with biotin, while 2G12 was conjugated with Simoa magnetic beads directly using the Quanterix homebrew kit (Quanterix, Bullerica, MA, USA). The automated Quanterix HD-1 analyzer was used as previously described [15,21]. MOLT IIIB cell lysates were prepared by lysing 1 × 10^6^ cells/mL in PBS containing 1% Triton X-100. Samples were stored at −80 °C until analysis. Upon thawing, cell lysates were serially diluted two-fold using 50% casein blocker in PBS (Thermo Fisher, Rockford, IL, USA) and 50% fetal bovine serum (FBS) (Thermo Fisher, Carlsbad, CA, USA). Samples were centrifuged at 20,000× *g* for 10 min at 4 °C to remove insoluble material prior to gp120 quantification using the Quanterix analyzer. All other assay reagents and reaction conditions followed the manufacturer’s protocol. gp120 concentrations were calculated based on a four-parameter calibration curve.

### 2.5. gp120 Liquid Chromatography–Mass Spectrometry (LC-MS)

For confirming the specificity of proteins measured by anti-gp120 antibodies and evaluating the detection sensitivity of LC-MS, immunoprecipitation (IP) followed by LC-MS was performed. MOLT IIIB cell lysates were generated by lysing 50 million cells in PBS containing 1% Triton X-100, followed by centrifugation at 20,000× *g* for 15 min. The anti-gp120 antibody VRC07 was conjugated to magnetic beads using the Dynabeads™ Antibody Coupling Kit (Cat# 14311D, Thermo Fisher Scientific Baltics UAB, Vilnius, Lithuania). Immunoprecipitation was performed by incubating lysates with antibody-conjugated beads overnight at 4 °C with end-to-end shaking. Captured proteins were eluted with 0.1% trifluoroacetic acid and further processed by neutralization (to pH~7.5), addition of internal standard, reduction with DTT (10 mM), alkylation with iodoacetamide (15 mM), and digestion with trypsin (1:20 enzyme to substrate) for 18h at 37 °C with shaking. The digestion was quenched by addition of acetic acid (3% final concentration). Qualitative analysis of the IP-captured material was performed using LC-MS with an Ultimate 3000 nanoflow LC coupled to a Thermo Orbitrap Fusion Lumos High-Resolution MS (Thermo Fisher Scientific, Waltham, MA, USA). Quantitative analysis of gp120 in samples was conducted using LC-MS with an Altis QQQ mass spectrometer coupled to a Waters M-class UPLC system (Milford, MA, USA). A stable-isotope-labeled synthetic peptide, IEPLGVAPTK, was used as an internal standard peptide for gp120 quantitation, and data were analyzed using Thermo Biopharma Finder v4.1 software [22].

### 2.6. gp120 Assay Pharmacological Validation

The novel gp120 quantitation assay was validated using the Jurkat cell line 2D10, developed in the laboratory of Jonathan Karn [13]. The 2D10 cell line, a latently infected Jurkat T-cell line, contains a lentiviral vector that expresses HIV-1 Env and the regulatory proteins Tat and Rev in *cis*, along with a short-lived GFP in place of Nef. Expression of the HIV-1 *env* gene encoding gp120 is suppressed under normal conditions but can be reactivated using stimulants such as TNFα, PMA/ionomycin, or the HDACi vorinostat. The 2D10 cell line was cultured in RPMI 1640 medium with L-glutamine (Gibco 11875), supplemented with 10% fetal bovine serum (Germini 100-106), 100 IU/mL penicillin, and 100 µg/mL streptomycin (Gibco 15140) and maintained under 5% CO_2_, 90% relative humidity, at 37 °C. PMA/ionomycin was applied at various concentrations to stimulate gp120 expression. Time- and dose-dependent gp120 expression following PMA/ionomycin stimulation was assessed. After treatment, 2D10 cells were pelleted by centrifugation at 400× *g* for 5 min. Pellets from 1 × 10^6^ cells were lysed with 1% Triton x-100 in 3% BSA/PBS lysis buffer, and lysates were stored at −80 °C until analysis. Prior to measurement, lysates were centrifuged at 20,000× *g* for 10 min at 4 °C, and supernatants were collected for gp120 quantification using ELISA. GFP expression in 2D10 cells was monitored using flow cytometry as mentioned above.

### 2.7. gp120 in HIV-1-Infected Human CD4^+^ T Cells Treated with Antiretrovirals

Peripheral blood mononuclear cells (PBMCs) were isolated from healthy donor blood collected in EDTA anticoagulant tubes. CD4^+^ T cells were purified from PBMCs using the EasySep human CD4^+^ T cell isolation kit (StemCell Technologies, Vancouver, BC, Canada) via negative selection following the manufacturer’s instructions. Cell counts and viability were determined using a Vi-Cell analyzer (Beckman Coulter, Brea, CA, USA). Purified CD4^+^ T cells (4 × 10^6^/mL) were activated for 72 h using Dynabeads^TM^ Human T-Activator anti-CD3/anti-CD28 beads (Gibco, Thermo Fisher Scientific Baltics UAB, Vilnius, Lithuania) at 25 μL per million CD4^+^ T cells. Activated CD4^+^ T cells were infected with HIV-1 virus (Vif-pNLG1-P2A) at a multiplicity of infection (MOI) of 0.5 for 4 h at 37 °C as previously described [20]. CD4^+^ T cells were then washed three times with cRPMI medium and centrifuged at 300× *g* for 5 min. Cells were resuspended at 5 × 10^6^ cells/mL in complete medium supplemented with 10 U/mL IL-2 and incubated for 12 h before treatment with 1 µM integrase-inhibitor raltegravir (RAL) to prevent second-round infection. HIV-1-infected CD4^+^ T cells were treated with final concentration of 100 nM TACK or non-TACK compounds [23] in the presence or absence of 250 nM indinavir (IDV). Cells were washed once and distributed into four treatment conditions: TACK alone, non-TACK alone, TACK with IDV, and non-TACK with IDV. Treatments were performed for 72 h. Following incubation, cells were collected to measure gp120 and p24 levels using ELISA, gp120-positive cells using ICC, and viral RNA levels using qPCR.

### 2.8. Quantitative Reverse-Transcription Polymerase Chain Reaction (qPCR)

The method for quantitating HIV viral RNA was adapted from a previously described protocol [24]. Total RNA was extracted from HIV-infected CD4^+^ T cells using the RNeasy kit (Qiagen, Hilden, Germany). qPCR was conducted using the TaqMan Fast Virus 1-Step Master Mix (Thermo Fisher Scientific, Carlsbad, CA, USA). Purified RNA (2 µL) was used as the template for each reaction. A custom-designed primers/probe set specific to HIV gag [20] was obtained from Thermo Fisher Scientific (Carlsbad, CA, USA). The housekeeping gene used as a reference, GAPDH, was also obtained from Thermo Fisher (Carlsbad, CA, USA). Amplification was performed using the QuantStudio 12K Flex system (Thermo Fisher, Carlsbad, CA, USA).

### 2.9. gp120-Immunocytochemistry (ICC)

A gp120 ICC method was developed based on modifications to a previously described HIV-1 p24 ICC protocol [20]. Following compound treatment, 0.5 million CD4^+^ T cells from 0.25 mL cultures were centrifuged and fixed onto microscopy slides. Staining was performed using a Leica Bond RX autostainer (Leica Biosystems, Nussloch, Germany) via a BOND Polymer Refine Red Detection kit (Leica Biosystems). HIV-1 gp120 protein was detected using an anti-gp120 monoclonal antibody (Cat# LS-C170976, LS Bio, Newark, CA, USA) (Table 1) at a final concentration of 3 µg/mL, diluted in Da Vinci Green Diluent buffer (Biocare Medical, Pacheco, CA, USA). Secondary detection employed an AP-labeled anti-mouse IgG detection antibody and AP red substrate was included in the kit. Staining procedures were automated according to the manufacturer’s instructions. After staining, slides were washed once in distilled water for 1 min, dehydrated twice with 100% EtOH for 5 min each, and immersed twice in HistoPrep xylene (Fisherbrand, Pittsburgh, PA, USA) for 5 min each. Slides were dried at room temperature for 20 min, mounted with EcoMount (Biocare Medical, Pacheco, CA, USA) and coverslips, and dried overnight at room temperature. Slides were scanned using a ZEISS Axioscan (Oberkochen, Germany) and images were analyzed using HALO digital pathology imaging software (v3.6.4134, Indica Labs, Albuquerque, NM, USA) [20,24].

### 2.10. Data Analysis

Statistical analyses were performed on log-transformed data, and results were reported after back transformation to the original scale. Data were presented as mean ± standard deviation. Graphs and figures were prepared using GraphPad Prism 11 (GraphPad Software, Inc., La Jolla, CA, USA). Statistical significance was assessed using Tukey–Kramer ANOVA or Student’s *t* test with thresholds indicated as *p* < 0.05 (*), *p* < 0.01 (**), and *p* < 0.001 (***).

## 3. Results

### 3.1. Development and Characterization of the gp120 Assay

An immunoassay for quantifying HIV-1 gp120 protein was developed by comparing sandwich ELISAs using either anti-gp120 antibodies or gp120’s native ligand, the CD4 protein. As shown in Figure 1, four distinct ELISA experiments were designed to develop a sensitive and specific gp120 immunoassay. First, the optimal gp120 detection reagent was determined by coating ELISA plate wells with either gp120 or gp160 protein and detecting with either sCD4-biotin plus streptavidin-AP or anti-gp120 antibody VRC07-AP. VRC07-AP produced a stronger signal compared to sCD4 protein detection for both recombinant HIV-1 glycoproteins (Figure 1A). These data indicate that the anti-gp120 antibody provides a more robust signal than the CD4 ligand. Second, the optimal capture reagent was determined using plates coated with either sCD4 protein or the anti-gp120 antibody 2G12, followed by detection with VRC07-AP. As illustrated in Figure 1B, capture using the 2G12 antibody consistently generated higher signals than sCD4 when detecting with both gp120 protein and MOLT IIIB cell lysate. Based on the combined results, a sandwich immunoassay using two anti-gp120 antibodies was selected for gp120 measurement. Next, to identify the best gp120 binding antibody, indirect ELISA was performed using 10 different anti-gp120 antibodies listed in Table 1. Seven antibodies demonstrated positive signals, with 2G12 producing the highest signal, followed by VRC07, Sigma anti-gp120 polyclonal antibody SAB3500463, and 447-52D (Figure 1C). Finally, the best antibody pair was determined by testing combinations of capture and detection antibodies. Three detection antibodies (2G12-AP, VRC07-AP, and 447-52D-AP) were tested with four capture antibodies (VRC07, 2G12, PG9, and Sigma anti-gp120 polyclonal antibody SAB3500463) using MOLT IIIB cell lysate as the source of the gp120 antigen (Figure 1D). The combination of VRC07 as the capture antibody and 2G12-AP as the detection antibody yielded the highest signal-to-background (S/B) ratio. Therefore, this antibody pair was selected for assay development and optimization. The assay’s characteristics were further evaluated (Figure 2). The standard curve for recombinant gp120 protein was linear across seven serial dilutions (10 pM to 0; R^2^ = 0.9999) (Figure 2A). The lower limit of quantitation (LLOQ) was determined to be 0.16 pM based on a coefficient of variation (%CV) < 20%. The assay demonstrated linearity in detecting gp120 from serial dilutions of MOLT IIIB cell lysate (Figure 2B). Specificity was confirmed by competitive inhibition with 100x unlabeled VRC07 or 2G12 antibodies, which reduced the signal by >90%, whereas the signal was unchanged with 100× unlabeled human IgG (hIgG) and no signal was detected in negative wells with BSA being substituted for lysate (Figure 2C). The assay also exhibited broad reactivity across HIV clades A, B, and C, as shown in Figure 2D. In summary, the developed novel gp120 ELISA demonstrated high sensitivity, specificity, and broad reactivity, making it suitable for sub-pM gp120 detection.

### 3.2. Comparison of Simoa and AP-ELISA Platforms for gp120 Measurement

To develop a more sensitive assay for gp120 quantitation using the selected 2G12 and VRC07 antibody pair, the single molecule array (Simoa), an ultrasensitive digital immunoassay platform, was evaluated and compared with the newly developed AP-ELISA. The Simoa platform demonstrated linear signal detection for recombinant gp120 across concentrations ranging from 50 pM to 0, with LLOQ reaching 0.23 pM. As shown in Figure 3A, Simoa successfully detected gp120 in MOLT IIIB cell lysates at a sample concentration equivalent to 500 cells per assay reaction. The signal decreased linearly following 2-fold serial dilutions of MOLT IIIB cell lysate, reaching a detection limit of approximately 31 cells per reaction, with a coefficient of variation (CV) of 18.3%. Simultaneously, gp120 was measured in MOLT IIIB cell lysate using the AP-ELISA platform (Figure 3B). The signal obtained at a concentration of 500 cells per reaction decreased linearly upon serial dilution to 16 cells. The detection limit for gp120 using AP-ELISA was also determined to be 31 cells per reaction, with a CV of 9.6%, matching the LLOQ observed with the Simoa platform. The correlation between gp120 measurements obtained with Simoa and AP-ELISA at identical cell concentrations was positive and exhibited a coefficient (R^2^) of 0.997 (*p* < 0.0001, n = 6), as shown in Figure 3C. In summary, the developed AP-ELISA demonstrated a detection limit comparable to the Simoa platform, confirming its sensitivity and reliability for gp120 quantitation using the same pair of anti-gp120 antibodies.

### 3.3. gp120 Immunoprecipitation and LC-MS

To further evaluate the specificity and sensitivity of the assay, immunoprecipitation (IP) using the anti-gp120 antibody VRC07 was performed on both recombinant gp120 protein (as a control) and MOLT IIIB cell lysate, followed by liquid chromatography–mass spectrometry (LC-MS) analysis. A gp120 peptide, spanning amino acids Ile496 to Lys505 (IEPLGVAPTK), was obtained after trypsin digestion of recombinant gp120 to act as the surrogate peptide for gp120 protein quantitation. A “heavy” isotope-labeled version of this peptide was synthesized and spiked into each IP sample at a constant concentration (1 nM) to serve as a reference standard for calculation. The ratio of LC-MS/MRM peak areas, representing the surrogate peptide (Ile496-Lys505) relative to the heavy-labeled reference peptide, was calculated as the assay readout. As shown in Figure 3D, the calculated ratio of the surrogate peptide to the heavy-labeled reference peptide decreased linearly with serial 10-fold dilutions of MOLT IIIB cell lysate containing 50,000 cells, 5000 cells, and 500 cells per sample. The ratio was 0.898, 0.103, and 0.028 across these dilutions, respectively. No significant change in the ratio was observed between 500 and 50 cells, indicating loss of quantitative accuracy below 500 cells. These results confirmed that the protein immunoprecipitated by VRC07 was HIV-1 gp120, and that the IP-LC-MS assay was able to detect HIV gp120 with an estimated limit of quantitation at 500 cells per sample (CV = 15.9%). Compared to the Simoa and AP-ELISA platforms, the LLOQ achievable by LC-MS/MRM was approximately 10 times higher, and, therefore, the AP-ELISA platform was selected for subsequent studies for its high throughput capability, sensitivity, specificity, and ease of use.

### 3.4. Validation of gp120 ELISA Using the 2D10 Cell Line

The gp120 ELISA, developed on the AP-ELISA platform, was validated using the 2D10 cell line, which expresses gp120 upon stimulation. As shown in Figure 4A, gp120 was not significantly detected in unstimulated 2D10 cells compared to the BSA negative control. However, robust gp120 expression was observed 24 h post-stimulation with either 100 ng/mL PMA and 1 µg/mL ionomycin or 5 ng/mL TNFα, and less pronounced expression was detected following treatment with 0.5 µM HDAC inhibitor (HDACi) vorinostat (VOR). The ranking of the amount of gp120 induction was PMA/ionomycin > TNFα > HDACi VOR. Based on these results, PMA/ionomycin was selected for further evaluation of gp120 expression in time-course and dose-response experiments. The time-course study, shown in Figure 4B, demonstrated that gp120 expression became significant at 12 h post-treatment compared to pre-stimulation levels. Peak gp120 expression was reached at 16 h and plateaued through 24 h post-induction. In the dose-response study, gp120 expression was evaluated 24 h post-PMA/ionomycin treatment. As shown in Figure 4C, gp120 levels in 2D10 cell lysate increased significantly with PMA concentrations ranging from 0.02 ng/mL to 20 ng/mL, demonstrating a dose-dependent response. The EC_50_ for PMA was calculated to be 0.52 ng/mL. Flow cytometry was also performed to monitor the expression of green fluorescent protein (GFP) in PMA/ionomycin-treated cells, alongside gp120 quantitation using the ELISA platform. As shown in Figure 4D, GFP expression correlated strongly with gp120 levels, with a correlation coefficient (R^2^) of 0.992 (*p* < 0.0001). Taken together, these results confirm that the developed gp120 ELISA is reliable for measuring gp120 expression in 2D10 cell lysates following stimulation with PMA/ionomycin and other agonists.

### 3.5. Measurement of gp120 in HIV-Infected CD4^+^ T Cells Treated with Antiretrovirals

To determine whether the gp120 protein assay can serve as a surrogate biomarker for the elimination of HIV-1-infected cells, gp120 levels were measured in lysates from in vitro HIV-1-infected human CD4^+^ T cells treated with 100 nM TACK molecule for 72 h. As shown in Figure 5A, gp120 was measured in lysates from HIV-1-infected CD4^+^ T cells in the control DMSO-treated group. A significant decrease in gp120 levels was observed in the TACK-treated group (*p* < 0.01, n = 3), with a 56% reduction in cell-associated gp120 compared to the DMSO group. No significant change in gp120 levels was detected in the TACK plus indinavir (IDV) group, where inhibition of HIV-1 protease by IDV abrogated the killing effect of TACK. Similarly, gp120 levels remained unchanged (*p* > 0.05) in cells treated with either non-TACK molecules or non-TACK plus IDV. Simultaneously, HIV-1 p24 protein levels were measured in lysates from the same samples. As shown in Figure 5B, the changes in p24 levels mirrored those of gp120. A significant reduction in p24 concentration was observed only in the TACK-treated group (*p* < 0.001), with no significant changes detected in the TACK and IDV co-treatment, non-TACK, or non-TACK and IDV co-treatment groups. The correlation between gp120 and p24 protein levels in the lysates was positive with a coefficient (R^2^) of 0.989 (*p* < 0.001) (Figure 5C). Cell-associated HIV-1 RNA was also quantified using qPCR targeting the HIV-1 gag region. As shown in Figure 5D, TACK treatment significantly reduced viral RNA levels (*p* < 0.001) compared to the DMSO-treated group. This effect was completely blocked by the addition of IDV, and no significant changes in viral RNA levels were observed in the non-TACK or non-TACK and IDV co-treatment groups. The correlation between gp120 levels and HIV-1 RNA levels yielded a correlation coefficient (R^2^) of 0.773 (*p* < 0.05) (Figure 5E). To evaluate the reduction in the number of HIV-1-positive cells caused by the treatment with TACK, gp120 ICC was developed using an anti-gp120 monoclonal antibody to stain gp120-positive cells. As shown in Figure 5F, approximately 8% of the CD4^+^ T cells were gp120-positive in the DMSO-treated group, reflecting an ~8% infection rate in the in vitro HIV-1 infection model used in this study. Following TACK treatment, the percentage of gp120-positive cells decreased significantly to ~3% (*p* < 0.001). This reduction was completely blocked by the addition of IDV, resulting in no significant change in gp120-positive cell numbers. Similarly, no major changes were observed in the non-TACK or non-TACK and IDV co-treatment groups, where the percentages of gp120-positive cells were 8.3% and 8.5%, respectively (*p* > 0.05). In summary, HIV-1 gp120, like HIV-1 p24 protein, can serve as a reliable surrogate biomarker for the killing of HIV-1-infected CD4^+^ T cells treated with the TACK molecule.

## 4. Discussion

In this study, a novel, sensitive, specific, and inexpensive sandwich ELISA for quantifying HIV-1 gp120 envelope glycoprotein was developed using two monoclonal antibodies, VRC07 and 2G12, known to bind distinct regions of gp120. The binding domain of 2G12 is exclusively located in the conserved V3 loop of gp120 and is dependent on N-linked glycan mannose, whereas VRC07 binds to the CD4-binding region of gp120 and exhibits broad activity against diverse HIV-1 strains, including clades A, B, and C [16]. The developed gp120 immunoassay demonstrated broad sensitivity across representatives from all three major HIV-1 viral clades, including 93RQ034 (clade A), JRFL B (clade B), and 93MW959 (clade C). This indicates that the assay can be broadly applicable for gp120 quantification across variant clades. Using the 2G12 and VRC07 antibody pair, the developed AP-ELISA platform demonstrated detection sensitivity and lower limits of quantitation (LLOQ) comparable to the ultrasensitive Simoa platform. However, this AP-ELISA is less expensive to conduct than Simoa and does not require specialized instrumentation. Both platforms exhibited greater than 10-fold improvement in detection sensitivity compared to previously reported traditional gp120 ELISAs [7,8,9,10]. Specificity for the gp120 protein was confirmed via immunoprecipitation with VRC07 and LC-MS analysis. The assay was validated with the 2D10 gp120-inducible cell line and demonstrated utility in pharmacological studies, including HIV-1-infected human CD4^+^ T cells treated with TACK. Together, these findings confirm that the newly developed gp120 assay is highly specific, suitable for quantification of gp120 in cell lysates, and has potential as a surrogate biomarker in cell-based killing studies.

Despite the success of combination antiretroviral therapy (cART) in transforming HIV-1 into a chronic manageable disease, cART cannot eradicate HIV-1 reservoirs, which persist in tissues and lead to viral rebound upon treatment cessation. Therefore, identifying novel biomarkers to quantify HIV-1 reservoirs prior to antiviral treatment interruption (ATI) is critical for advancing HIV-1 eradication strategies and achieving functional cure. Several biomarkers, including intact and defective integrated HIV-1 DNA, viral RNA, and HIV p24 protein, have been studied in the context of HIV-1 cure research [25,26,27,28,29]. However, these biomarkers are imperfect as most HIV-1 DNA is defective, not all intact RNA is translated into protein, and HIV-1 p24 protein can originate from defective viruses [30]. Additionally, p24 expression is extremely low in lymphatic tissues and peripheral blood cells, necessitating the exploration of alternative biomarkers for detecting both HIV-1-infected cells and viral replication. Plasma gp120 has previously been used as a diagnostic biomarker for HIV-1 infection [7,8]. More recently, gp120 has emerged as an important therapeutic target in ADCC studies due to its presence on the surface of infected CD4^+^ T cells and is currently being exploited in non-human primate (NHP) models and clinical trials for various HIV-1-killing therapies. Thus, gp120 has potential as a biomarker for HIV-1 killing, in addition to the HIV-1 p24 protein which has been used previously [15,20]. In the present study, we also showed that gp120 protein levels in lysates from HIV-1-infected CD4^+^ T cells decreased significantly after being treated with TACK while remaining unchanged in the control groups in the presence or absence of the protease inhibitor IDV. TACK compounds are unique dual-functioning molecules that both inhibit HIV-1 reverse transcriptase (RT) and induce premature activation of HIV-1 protease, leading to CARD8-mediated inflammasome activation and pyroptosis in infected cells [31]. This mechanism targets translationally active HIV-1 reservoirs, resulting in significant reductions of gp120 levels in infected cell lysates. We demonstrated that gp120 levels correlated well with HIV-1 p24 protein and viral RNA levels, further supporting its utility as a biomarker for assessing the killing potency of TACK molecules in cell-based assays. Importantly, gp120 provides an orthogonal approach to confirm translational integrity of HIV-1-infected reservoir cells that may contain defects in p24 levels. Further research is needed to evaluate whether gp120 expression changes in infected primary cells from people with HIV-1 (PWH) following treatment with latency-reversing agents (LRAs) such as PMA and to investigate its utility in in vitro and in vivo cure studies.

Expression of HIV-1 gp120 is extremely low in infected host cells during latency, both in lymphatic tissues and peripheral blood. Improving assay sensitivity is therefore essential for addressing diagnostic and therapeutic challenges. Several platforms have been reported to achieve ultra-high detection sensitivity, including Simoa digital ELISA, S-plex Meso-Scale Discovery (MSD) [32,33], NULISA [34], and AP-ELISA [12,19]. In this study, the AP-ELISA platform was compared to Simoa using the same pair of anti-gp120 antibodies (2G12 and VRC07). Both platforms demonstrated comparable sensitivity, with a LLOQ of 31 MOLT IIIB cells per reaction. Based on these findings, AP-ELISA was selected for gp120 quantification in studies using TACK due to its ease of use, high throughput, and affordability. Although the S-plex MSD platform was previously evaluated in-house for measuring HIV-1 p24 protein, its sensitivity was lower than that of Simoa or AP-ELISA. NULISA has been reported to achieve significantly higher sensitivity and shows promise for improving immunoassay detection limits [35]. However, NULISA was not included in this study due to lack of availability in-house. Taken together, the developed AP-ELISA for gp120 quantification is a novel, sensitive, and specific assay which is simple, cost-effective, and well-suited for measuring gp120 in HIV-1-infected cells. It provides a valuable tool for HIV-1-killing studies and holds potential for further applications in HIV-1 cure research.

## 5. Conclusions

A novel, sensitive, and specific protein quantification assay for HIV-1 envelope gp120 was developed and validated pharmacologically in both a 2D10 envelope inducible cell line and in HIV-1-infected cell-based killing assays utilizing the TACK mechanism of action. The quantification of gp120 protein in cell lysate offers a potential biomarker, alongside p24 and viral RNA, for evaluating HIV-1-infected cells. Further studies are required to validate this biomarker in infected primary cells taken from people with HIV (PWH) following treatment with LRAs.

## Figures and Tables

**Figure 1 viruses-18-00046-f001:**
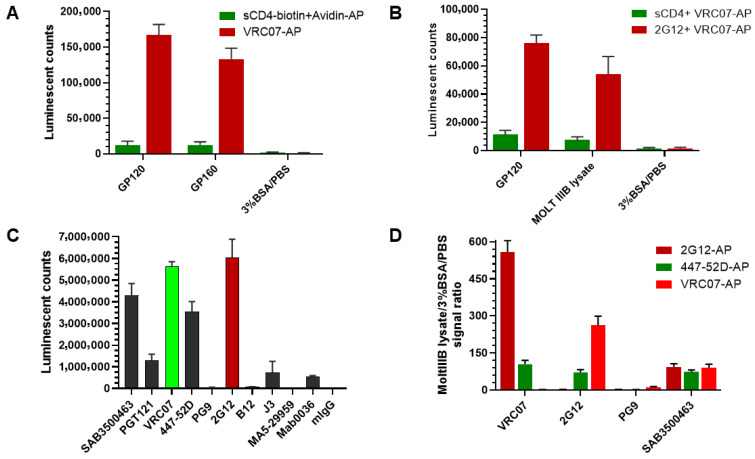
Selection of the capture and detection antibody pair for gp120 assay development. (**A**) Anti-gp120 antibody VRC07 superiority to gp120 ligand CD4 protein in detecting the coated gp120 protein. (**B**) Anti-gp120 antibody 2G12 superiority to CD4 protein in capturing gp120 protein. (**C**) Comparison of anti-gp120 antibodies in indirect ELISA. (**D**) Comparison of different antibody pairs for detecting gp120 in sandwich ELISA.

**Figure 2 viruses-18-00046-f002:**
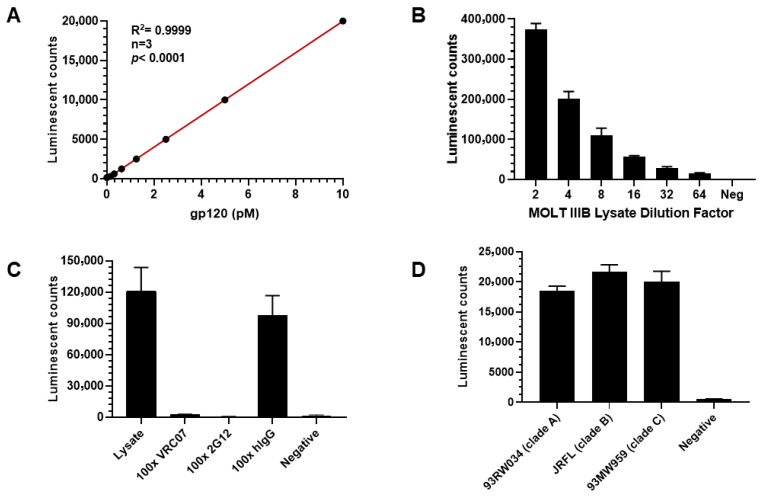
gp120 assay characterization and validation. (**A**) gp120 assay standard curve in sandwich ELISA using VRC07 antibody capture and alkaline phosphatase (AP)-labeled anti-gp120 antibody 2G12 for detection. (**B**) MOLT IIIB cell lysate linear decline in gp120 signal when diluted from neat to 1: 64-fold. (**C**) Specificity of signal measured using 100-fold unlabeled capture 2G12 or detection VRC07 antibody. Addition of 100-fold unlabeled 2G12 or VRC07 antibody led to an almost complete loss of gp120 signal. (**D**) Developed gp120 ELISA assay detecting gp120 sequence from HIV clades A, B, and C, respectively.

**Figure 3 viruses-18-00046-f003:**
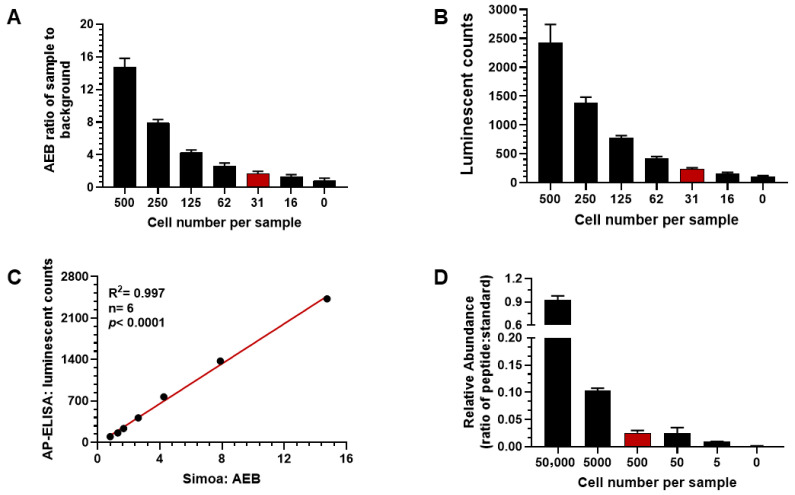
Comparison of gp120 measurement assay platforms. (**A**) Average enzymes per bead (AEB) measured with single molecule array (Simoa) in Quanterix declining linearly from 500 to 16 cells per reaction serial dilution. LLOQ = 31 MOLT IIIB cell lysate per reaction. (**B**) Comparison of the luminescent signal obtained from AP-ELISA following MOLT IIIB lysate dilution with LLOQ = 31 cells per reaction. (**C**) Correlation in signal between Simoa and AP-ELISA platforms (r = 0.998, n = 7, *p* < 0.0001). (**D**) LC/MS/MRM confirming the specificity of gp120 immunoprecipitated by anti-gp120 antibody VRC07. Signal obtained from 50,000 MOLT IIIB cells lysate decreased linearly at 10-fold dilution to 500 cells; signal plateaued with LLOQ = 500 cells per reaction.

**Figure 4 viruses-18-00046-f004:**
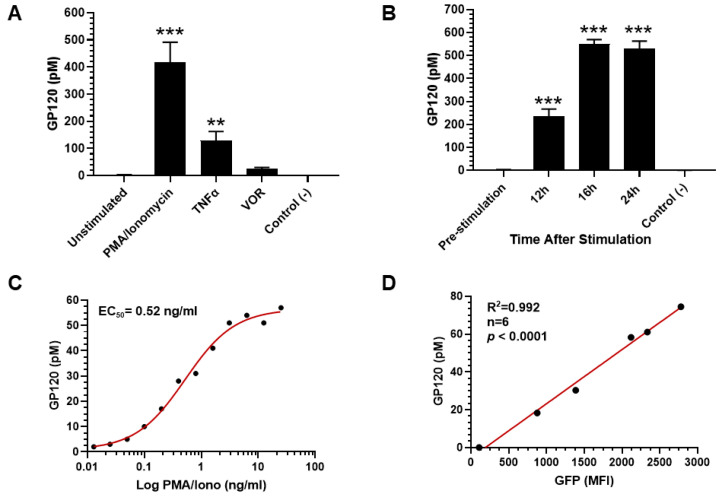
Pharmacological validation of gp120 assay in 2D10 cell line. (**A**) gp120 concentration increasing significantly in 2D10 cell lysate at 24 h following latency reversing agents PMA/ionomycin, TNFα, or HDACi VOR treatment. (**B**) Time course of gp120 expression in 2D10 cell line. Peak expression occurring at 16 h post-treatment. (**C**) Dose response observed at 24 h following PMA/ionomycin treatment with the EC_50_ = 0.52 ng/mL. (**D**) Detection of green fluorescent protein (GFP) tagged gp120 correlating with the expression of gp120 (r = 0.996, n = 6, *p* < 0.0001). ** *p*< 0.01; *** *p*< 0.001.

**Figure 5 viruses-18-00046-f005:**
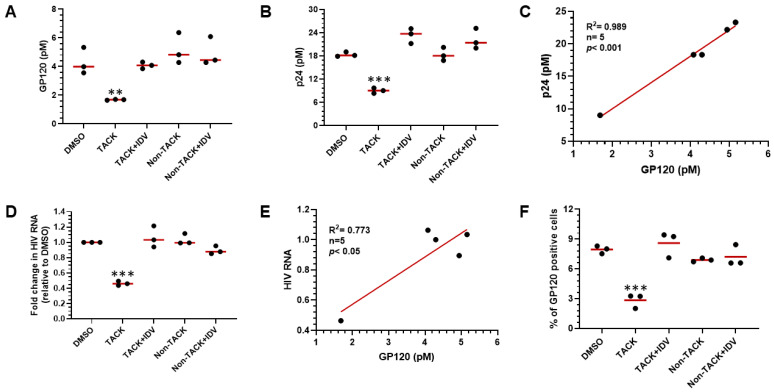
Application of gp120 detection to quantification of TACK activity. (**A**) Quantification of HIV-1 gp120 concentration by AP-ELISA. (**B**) Quantitation of HIV-1 p24 level by Simoa. (**C**) Correlation between HIV-1 gp120 and p24 markers (r = 0.994, n = 5, *p* < 0.001). (**D**) Quantification of cell-associated HIV-1 RNA by qPCR. (**E**) Correlation between gp120 and HIV-1 viral RNA markers (r = 0.879, n = 5, *p* < 0.05). (**F**) Quantification of gp120-positive CD4^+^ T cells by ICC. Non-TACK: TACK inactive NNRTI; IDV: HIV protease inhibitor indinavir. ** *p*< 0.01; *** *p*< 0.001.

**Table 1 viruses-18-00046-t001:** List of anti-gp120 monoclonal and polyclonal antibodies.

Reagents	Epitope	Catalog#	Manufacturer	City, State, Country
anti-gp120 (PGT-121)	V3 loop glycan	PABZ-203	Creative Biolabs	Shirley, NY, USA
anti-gp120 (VRC07)	CD4 bs*	N/A	In-house	San Francisco, CA, USA
anti-gp120 (447-52D)	V3 loop	AB014	Polymun Scientific	Klosterneuburg, Austria
anti-gp120 (PG9)	V1/V2 loop	MRO-3462CQ	Creative Biolabs	Shirley, NY, USA
anti-gp120 (2G12)	V3 loop glycan	PABC-062	Creative Diagnostics	Shirley, NY, USA
anti-gp120	CD4 bs	B12	Absolute antibody	Cleveland, UK
anti-gp120 (J3)	CD4 bs	MRO-3503CQ	Creative Biolabs	Shirley, NY, USA
anti-gp120	unknown	MA5-29959	ThermoFisher	Rockford, IL, USA
anti-gp120	unknown	Mab0036-M05	Abnova	Taoyuan City, Taiwan
anti-gp120	unknown	LS-C170976	LS Bio	Newark, CA, USA
Poly anti-gp120	aa497-511	SAB3500463	Millipore Sigma	St. Louis, MO, USA

bs*: binding site.

## Data Availability

The data presented within this study are available upon request to the corresponding authors.

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
