# Peer review of "Development of an Ultrasensitive ELISA Assay for Evaluating HIV-1 Envelope Glycoprotein as a Marker for Targeted Activator of Cell Kill"

_viruses, 2025, doi:10.3390/v18010046_

Round 1
Reviewer 1 Report
Comments and Suggestions for Authors
The authors have developed a novel, highly sensitive and specific protein quantification assay for detecting gp120, a HIV-1 envelope. The methods are well described and the experimental design is highly thought out.
The sensitivity of this assay is demonstrated clearly. However, I wished the authors have investigated HIV-1 infected cells from patients under antiretroviral therapy. The sensitivity of this assay would be more reliable showing positive results from patient samples. Nevertheless, the authors have stated in their manuscript that these experiments haven’t been carried out and future studies with patient samples will are needed.
Reviewer 2 Report
Comments and Suggestions for Authors
This manuscript presents a case of a good experimental design and implementation, however, the description of goals and results leaves some questions.
- The title is unclear. It seems to dictate a certain addendum, e.g. "Evaluating HIV-1 envelope glycoprotein as a marker for targeted activator of cell kill [activity, or efficacy of action, etc.]". Moreover, the main emphasis is made on the development of an ultrasensitive sandwich ELISA method, and not to the theme posted as a title. It is especially notable in the Abstract , which should be revised according to the declared title.
- The main emphasis is made on the development of a new sensitive ELISA assay for gp120 protein, but there is no explanation, why there has been a need in developing this assay. Was it necessary only to increase the sensitivity via using a pair of efficiently binding antibodies? Or a simpler method with a comparable sensitivity was required? If the question was in the sensitivity, there should be a comparison with another method, which authors consider a less sensitive one. Was the Simoa approach for gp120 detection developed earlier by the same authors or in parallel with the sandwich ELISA?
- In order to select proper antibodies for a sandwich ELISA for gp120, one should know their at least approximate B-epitope specificity. If there is an information about it, this information should be mentioned in the Table 1, otherwise the selection of certain antibodies remains unclear.
- There is an ethical concern about this study. Donor blood has been used for CD4+ cell preparation. What was the source of this material? Was the informed consent obtained from the donors?
- The peptide IEPLGVAPTK (496-505) is not a "trypsin-digested peptide" (line 312), but a gp120 characteristic fragment obtained after trypsin digestion.
Round 2
Reviewer 2 Report
Comments and Suggestions for Authors
the authors have made the title of the manuscript closer to he presented results and their discussion. Also the text has been improved in orer to make the authors' idea, results and conclusions more clear, no ethical concerns are left as well. I have no other comments or notes.